# Cold deep subduction recorded by remnants of a Paleoproterozoic carbonated slab

Cheng Xu[1], Jindřich Kynický[2,3], Wenlei Song[2,3], Renbiao Tao[1,4], Zeng Lü[1], Yunxiu Li[1], Yueheng Yang[5], Miroslav Pohanka [2,3], Michaela V. Galiova[2,3], Lifei Zhang [1] & Yingwei Fei [1,4]

The absence of low-thermal gradients in old metamorphic rocks (<350 °C GPa$^{-1}$) has been used to argue for a fundamental change in the style of plate tectonics during the Neoproterozoic Era. Here, we report data from an eclogite xenolith in Paleoproterozoic carbonatite in the North China craton that argues for cold subduction as early as 1.8 Ga. The carbonatite has a sediment-derived C isotope signature and enriched initial Sr–Nd isotope composition, indicative of ocean-crust components in the source. The eclogite records peak metamorphic pressures of 2.5–2.8 GPa at 650–670 °C, indicating a cold thermal gradient, 250(±15) °C GPa$^{-1}$. Our data, combined with old low-temperature events in the West African and North American cratons, reveal a global pattern that modern-style subduction may have been established during the Paleoproterozoic Era. Paleoproterozoic carbonatites are closely associated with granulites and eclogites in orogens worldwide, playing a critical role in the Columbia supercontinent amalgamation and deep carbon cycle through time.

---

[1] Key Laboratory of Orogenic Belts and Crustal Evolution, School of Earth and Space Sciences, Peking University, Beijing 100871, China. [2] Department of Geology and Pedology, Mendel University, Brno 61300, Czech Republic. [3] Central European Institute of Technology, Brno University of Technology, Brno 61600, Czech Republic. [4] Geophysical Laboratory, Carnegie Institution of Washington, 5251 Broad Branch Road NW, Washington, DC 20015, USA. [5] Institute of Geology and Geophysics, Chinese Academy of Sciences, Beijing 100029, China. Correspondence and requests for materials should be addressed to C.X. (email: xucheng1999@pku.edu.cn) or to L.Z. (email: lfzhang@pku.edu.cn) or to Y.F. (email: yfei@carnegiescience.edu)

Plate tectonics has reshaped the distribution of the Earth's continental and oceanic lithosphere through time. However, when modern-style plate tectonics started is still debated, with proposals ranging from the Neoproterozoic Era to the Archean Eon[1-7]. The absence from the geological record of high-pressure and low-temperature metamorphism, including blueschists and ophiolites before the Neoproterozoic Era, presents a challenge to understanding early subduction. Archean mantle is about 300 K hotter than the present mantle and high mantle temperature would have significant effects on the viability and style of the subduction process[8]. As a result, early ocean plates may have been more buoyant and richer in MgO but weaker than modern ocean lithosphere. The Mg-rich composition of the Archean oceanic crusts may preclude blueschist formation[9]. Popular models[10,11] for early plate tectonics involve shallow flat subduction and delamination of thick crust, precluding low thermal gradient deep subduction. This interpretation is supported by thermodynamic modeling of metamorphic mineral assemblages that the Archean tonalitic–trondhjemitic–granodioritic (TTG) compositions may be formed by shallow subduction of thick oceanic crust[12]. The petrological record also shows that the trace element signatures in TTG magmas can only be produced when melts separate from a garnet amphibolite residue, not an eclogite residue[13]. However, low Paleoproterozoic thermal gradients are recorded in blueschist facies metamorphism (~350 °C GPa$^{-1}$) in West Africa[3] and retrograded amphibolitized eclogite boudins (~300 °C GPa$^{-1}$) in North America[6]. These recent findings challenge the timing of onset of modern-style subduction.

A global record of low geothermal gradients in the Paleoproterozoic is a prerequisite to demonstrate a change of plate tectonic processes that dates back to that time. During the Paleoproterozoic Era, there were global-scale continent and mountain building events from West Africa to North America and to North China, leading to Columbia, the first coherent supercontinent in the Earth history[14,15]. The North China craton (NCC, Fig. 1a) is one of the fundamental Precambrian nuclei of Asia and one of the oldest cratonic blocks in the world (~3.8 Ga)[16]. It preserves a record of a long and complex crustal evolution, but evidence of a thick mantle keel beneath Archean-aged crust is absent[17]. The Eastern and Western blocks evolved independently during the Archean and collided along the Trans-North China Orogen (TNCO, Fig. 1b) during the Paleoproterozoic (~1.85 Ga) to form the NCC[16]. The appearance of Paleoproterozoic carbonatites in the NCC offers an opportunity to evaluate subduction–accretion collision tectonics during the Paleoproterozoic Era. Here, we report pressure–temperature (P–T) conditions retrieved from an eclogite xenolith in a Paleoproterozoic carbonatite that provides direct petrological evidence of cold oceanic subduction during the Paleoproterozoic Era.

## Results

**Geological background.** The carbonatites are associated with pyroxenites, pyroxene syenites, and occasionally with granulites (Supplementary Fig. 1a, b). They occur as dykes and were emplaced in the Fengzhen and Huai'an areas at the crossing between the Khondalite belt and TNCO (Fig. 1b). These orogens are Paleoproterozoic collisional belts formed by the amalgamation of the Western and Eastern blocks and before that the Ordos and Yinshan blocks, respectively[16]. The Khondalite belt is composed of Paleoproterozoic sillimanite–garnet and garnet–biotite gneisses. In the TNCO, late Archean crust is dominated by TTG gneisses. Paleoproterozoic mafic granulites are widely distributed (Fig. 1b). Their P–T paths are mostly clockwise, related to collision/exhumation processes, and some of them evolved through eclogite facies[17]. The granulites, occurring as lenses and tabular

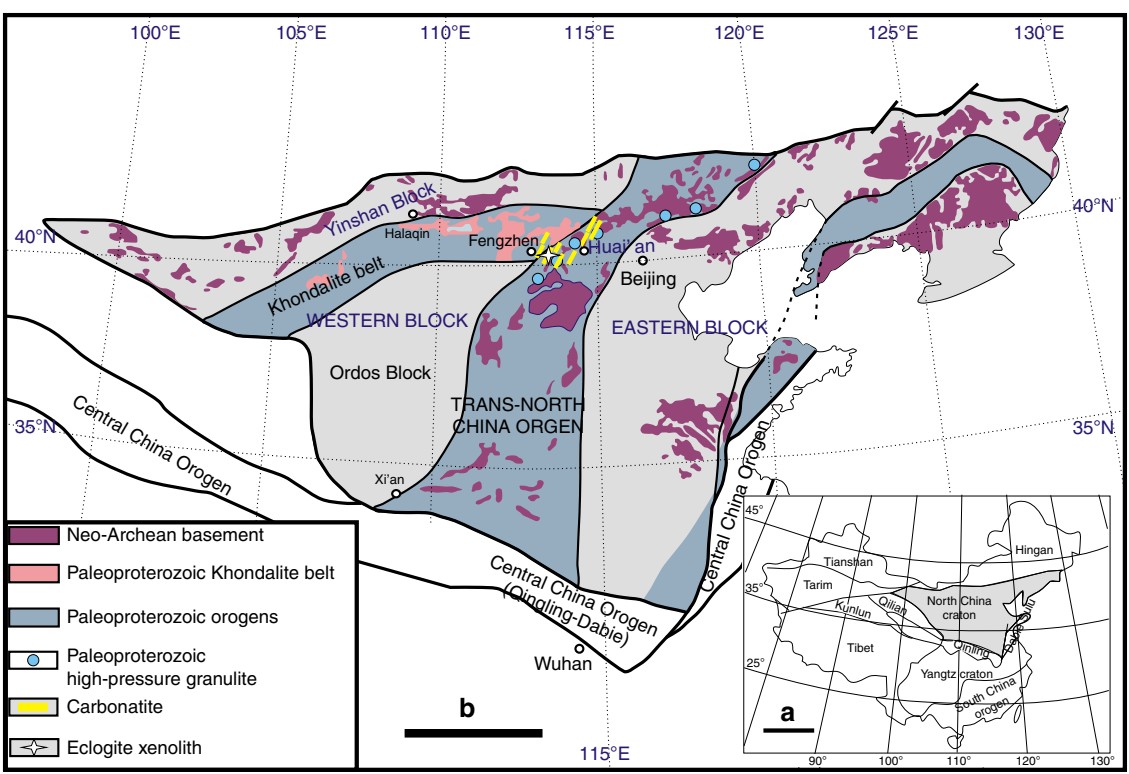

**Fig. 1** Schematic geological map of the North China craton (NCC) with the location of the eclogite xenoliths. The maps are after Zhao et al.[16]. **a** The NCC is separated from Yangtze craton by Qinling orogen[16]. **b** The Eastern and Western blocks in the NCC were amalgamated by the Paleoproterozoic Trans-North China Orogen (TNCO)[16]. The Paleoproterozoic carbonatite and its associated eclogite xenoliths and high-pressure granulites[17] occur in the TNCO. The scale bar is 800 and 400 km in **a** and **b**, respectively

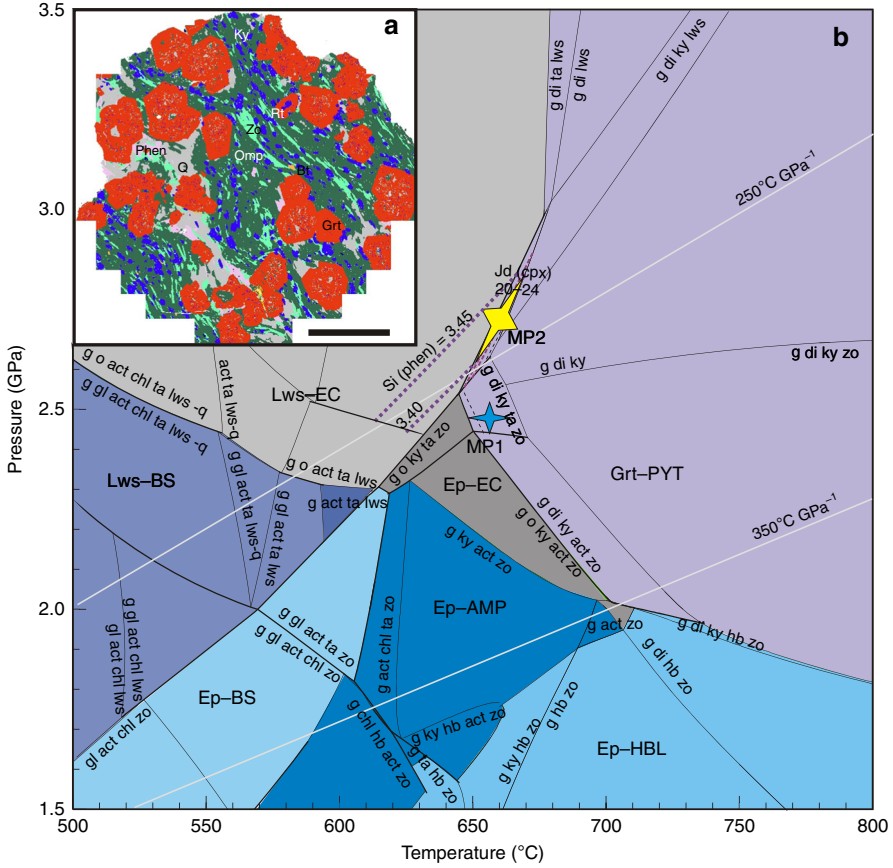

**Fig. 2** Mineral mapping of the fresh eclogite xenolith and *P–T* phase diagram. **a** The eclogite contains typical minerals of garnet (Grt) and omphacite (Omp), which do not show any retrogressed feature. The coarse-grained garnets are rimmed by quartz (Q) and phengite (Phen), indicating a relatively high pressure and low temperature metamorphism. Ky kyanite, Zo zoisite, Bt biotite, Rt rutile. The scale bar is 5 mm. **b** *P–T* diagram is calculated in the NCKFMASH system[27] according to the effective bulk-rock composition obtained from **a**. The peak pressure and temperature (labeled as MP1) are calculated using the Zr-in-rutile and mineral-pair thermobarometer (Supplementary Table 7). The peak metamorphic pressure and temperature, labeled as MP2, are independently constrained by the jadeite and Si contents of omphacite and phengite (purple dash line), respectively. The standard deviation (±1.4 kbar) in pressure is adopted on the basis of a Monte Carlo assessment on the phase equilibrium modeling[53]. Combining MP1 and MP2 data gives a thermal gradient of 250(±15) °C GPa$^{-1}$. Lws lawsonite, Ep epidote, Grt garnet, Jd jadeite, CPX clinopyroxene, Phen phengite, EC eclogite, PYT pyroxenite, BS blueschist, AMP amphibolite, HBL hornblendite

bodies associated with the carbonatites, consist of plagioclase, clinopyroxene, orthopyroxene, amphibole, garnet, and quartz. They are characterized by augen structures consisting of plagioclase and orthopyroxene enveloped by porphyroblastic garnet, indicating rapid exhumation (Supplementary Fig. 1c). Guo et al.[18] obtained an age of ~1.82 Ga from the Huai'an granulite, with peak *P–T* conditions of 750–850 °C and 1.1–1.5 GPa.

**Mineralogy and geochronology.** The carbonatite is mainly composed of calcite (70–80%), with minor amounts of diopside, apatite, phlogopite, olivine, and spinel (Supplementary Fig. 2 and Supplementary Table 1). Apatite occurs as well-developed elongate prismatic crystals (typically ~5 mm in length). In-situ LA-ICPMS U–Pb dating of apatite yields weighted U–Pb ages of 1681 (±61) Ma (Fengzhen area) and 1765(±35) Ma (Huai'an area) (Supplementary Fig. 2 and Supplementary Table 2).

Within the Paleoproterozoic carbonatites, there are rare but well-preserved cm-sized eclogite xenoliths (Supplementary Fig. 1d). The eclogite is composed of euhedral garnet porphyroblast up to 1 cm in diameter (~37 vol%), omphacite (~36 vol%), kyanite (~9 vol%), quartz (~9 vol%), zoisite (~7 vol%), and phengite (~2 vol%), with accessory amphibole, biotite, and rutile (Supplementary Fig. 3 and Supplementary Table 3). The mineral

proportions were obtained from a SEM–EDS area-scan of one eclogite mount (Fig. 2a). Garnet has elevated Si (3.019 ± 0.005 pfu on the base of 12 oxygen atoms) and compositional traverses reveal prograde concentric zoning with a homogeneous core overgrown by a rim of increasing pyrope but decreasing almandine, showing a Mg/(Mg + Fe) ratio increasing from the core to the rim (Supplementary Fig. 4). The garnet core traps abundant mineral inclusions, including kyanite, quartz, zoisite, omphacite, Ca–Na amphibole, Na amphibole, paragonite, and rutile, whereas the rim is relatively free of inclusions (Fig. 2a). The jadeite content of omphacite in the matrix and inclusions is 23 (±1) and 27(±3) mol%, respectively. Zoisite is poor in $Fe^{3+}$. Subparallel zoisite grains in the matrix define a foliation that is characteristic of orogenic eclogite (Fig. 2a). Phengite has high Si contents (3.40–3.44 pfu). Large matrix phengite is texturally equilibrated with garnet and omphacite, and contains omphacite and kyanite inclusions, interpreted as peak metamorphic minerals. Minor magnesiohornblende occurs in the matrix, and has higher MgO (~20 wt%) and lower $Na_2O$ (~1.5 wt%) contents than inclusions in the garnet core. The Ca–Na amphibole in garnet is barroisite with elevated Na ($Na_B = 0.5–0.7$ pfu) and is rarely associated with paragonite. Rare sodic amphibole in garnet contains high Mg and Na, and low Ca (~4.1, 1.2, and 0.01 pfu, respectively). Minor monazite occurs in association with calcite

veinlets that crosscut the garnets and primary minerals in the garnet cores. In-situ electron microprobe dating of two groups of monazites (primary and late-stage growth) yields different weighted U–Th–Pb ages of 1839(±26) and 1766(±7) Ma (Supplementary Fig. 5 and Supplementary Table 4). These ages constrain the eclogite metamorphism to the Paleoproterozoic Era, similar to the Huai'an granulite age in the TNCO[18].

**Peak metamorphic $P$ and $T$ conditions of eclogites**. We derived the bulk rock composition of the eclogite xenolith by integrating the mineral compositions from electron microprobe analysis and the modes acquired by the SEM–EDS area-scan (Supplementary Table 5). The eclogite has a similar composition to oceanic gabbro (Supplementary Fig. 6), but is different from the eclogite xenoliths in kimberlites[19]. The latter contains inclusion-free garnet and clinopyroxene, and no quartz. Based on in-situ mineral trace element analyses in the eclogite xenolith (Supplementary Table 6), the garnets have a strong depletion in the light rare earth elements (LREEs) relative to the heavy (H)REEs, and show a nearly flat chondrite-normalized (cn) HREE pattern with $Y/Yb_{cn}$ of ~1 (Supplementary Fig. 7). In contrast, zoisite contains high LREE contents with a $La/Yb_{cn}$ of ~266, and is characterized by a positive Eu anomaly. According to these mineral compositions and modal mass, the reconstructed whole-rock shows slightly depleted LREEs ($La/Yb_{cn}$ = 0.7), flat HREE ($Y/Yb_{cn}$ = 1), and positive Eu anomaly. The main hosts for the incompatible trace elements are zoisite [LREEs, Sr (3574–3988 ppm), Th (2–10 ppm), U (1–2 ppm), Pb (10–11 ppm)], garnet [Y (13–31 ppm), HREEs], and phengite [Rb (183–201 ppm), Ba (2428–2564 ppm)]. It is suggested that these trace-element budgets in eclogites are controlled by newly formed minerals such as zoisite, garnet, and phengite, during prograde metamorphism in subduction zones[20].

To estimate the peak metamorphic conditions, we first use the Zr-in-rutile thermometer to determine the peak temperature. The Zr concentration in the matrix rutile ranges from 150 to 174 ppm (Supplementary Table 7). The Zr-in-rutile thermometer[21] yields temperatures of 650 and 670 °C for the minimum and maximum Zr concentrations, respectively (Supplementary Table 8). For the temperatures of 650 and 670 °C, the garnet–clinopyroxene–phengite (Grt–Cpx–Phen) barometer[22] gives peak pressures of 2.49 and 2.46 GPa, respectively (Supplementary Table 8). The Grt–Cpx–Phen barometer of ref. [22] is preferred because of the low jadeite content (<0.5) of omphacite in the mineral assemblage[23]. Using the pressure-dependent Zr-in-rutile thermometer[24], the calculated temperature is 658 and 669 °C at 2.5 GPa for the minimum and maximum Zr concentrations, respectively (Supplementary Table 8), consistent with the result from the Grt–Cpx–Phen barometer. Further confirmation of the temperature determination comes from the garnet–clinopyroxene (Grt–Cpx) thermometer. Because grossular in the eclogitic garnet is less than 0.35, we use the formula of Powell[25] as suggested by ref. [26]. The compositions of garnet rim and omphacite in the matrix yield a mean temperature of 670 °C at 2.5 GPa.

Multiple thermobarometers give consistent estimates of peak pressures (2.5 GPa) and temperatures (650–670 °C). For comparison, Fig. 2b shows a $P$–$T$ phase diagram calculated for the eclogite in the simplified system $Na_2O$–$CaO$–$K_2O$–$FeO$–$MgO$–$Al_2O_3$–$SiO_2$–$H_2O$ (NCKFMASH) with excess water, using the program THERMOCALC[27]. The peak assemblage of garnet + omphacite + kyanite is stable in the $P$–$T$ range of 2.6–2.8 GPa and 650–670 °C (Fig. 2b), based on isopleths of jadeite content in omphacite (22–24) and Si content in phengite (3.40–3.44). This is consistent with the thermobarometric estimates

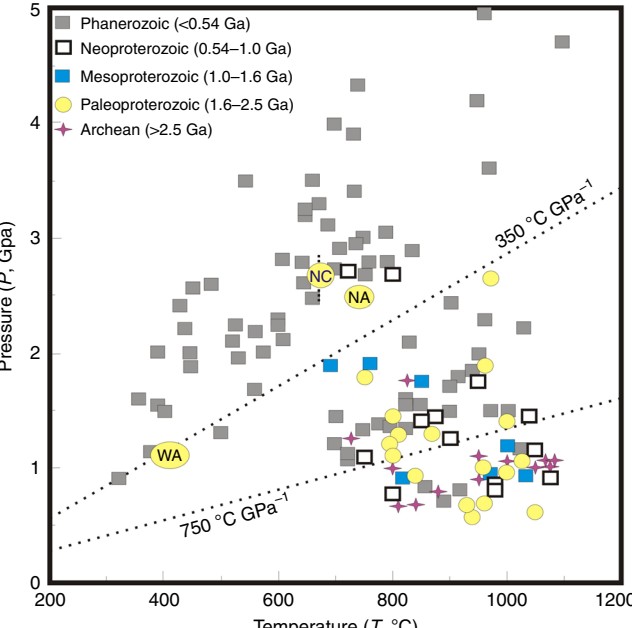

**Fig. 3** Peak pressure–temperature condition of this study compared to a global compilation of metamorphic belts. The $P$–$T$–$t$ data of these metamorphic belts are indicated by different symbols (see legend) and grouped into three types of metamorphism[2]: high- and ultrahigh-pressure metamorphism (<350 °C GPa$^{-1}$); eclogite and high-pressure granulite metamorphism (<750 °C GPa$^{-1}$); and granulite and ultrahigh-temperature metamorphism (>750 °C GPa$^{-1}$). A Paleoproterozoic thermal gradient of <350 °C GPa$^{-1}$ was recorded in the high-pressure metamorphism from North China (NC), West Africa (WA)[3], and North America (NA)[6] cratons

summarized above. These peak conditions correspond to a low apparent thermal gradient of 250(±15) °C GPa$^{-1}$ (Fig. 3). This is among the coldest subduction recorded in the Paleoproterozoic Era and akin to many Phanerozoic blueschist terranes exhumed from oceanic subduction zones (<350 °C GPa$^{-1}$)[2] (Fig. 3).

**Isotope geochemistry of carbonatites**. To understand the sources of carbon, we measured the oxygen and carbon isotopic compositions of the carbonatites (Supplementary Table 9). The $\delta^{18}O_{V-SMOW}$ and $\delta^{13}C_{V-PDB}$ values (9.4–15.8‰ and −5.7 to −1.6‰, respectively) are higher, compared to those of normal mantle rocks (5–8‰ and −7 to −5‰, respectively) and typical mantle-derived primary carbonatites (6–10‰ and −8 to −4‰, respectively)[28] (Fig. 4a). While the oxygen isotopic values may be readily modified, the carbon isotopic values are much less susceptible to later processes like fluid infiltration and wall–rock assimilation. The high $\delta^{13}C$ values recorded in the carbonatites potentially can be accounted for by fractional crystallization, or assimilation of sediments or subducted oceanic crust[28]. Fractional crystallization is unlikely to have been the cause for the high $\delta^{13}C$ values because there is no defined trend between the $\delta^{13}C_{V-PDB}$ and $\delta^{18}O_{V-SMOW}$ values. Similarly, only minimal wall–rock assimilation is possible based on a negative Pb anomaly documented in the whole rocks[29]. In contrast, heavy-C enrichment in the carbonatites can be easily explained by the involvement of recycled oceanic carbonates, with typical $\delta^{13}C_{V-PDB}$ values of 0 (±1.5)‰[30] (Fig. 4a), associated with subducted oceanic crust.

The isotopic characteristics of Sr and Nd in carbonatites are generally inherited from their source regions[28]. In this case, the carbonatites and their apatites show very unusual negative $\varepsilon_{Nd}(t)$ values (−5.8 to −7.8) (Supplementary Table 10), indicating the involvement of a substantial amount of ocean crustal materials in

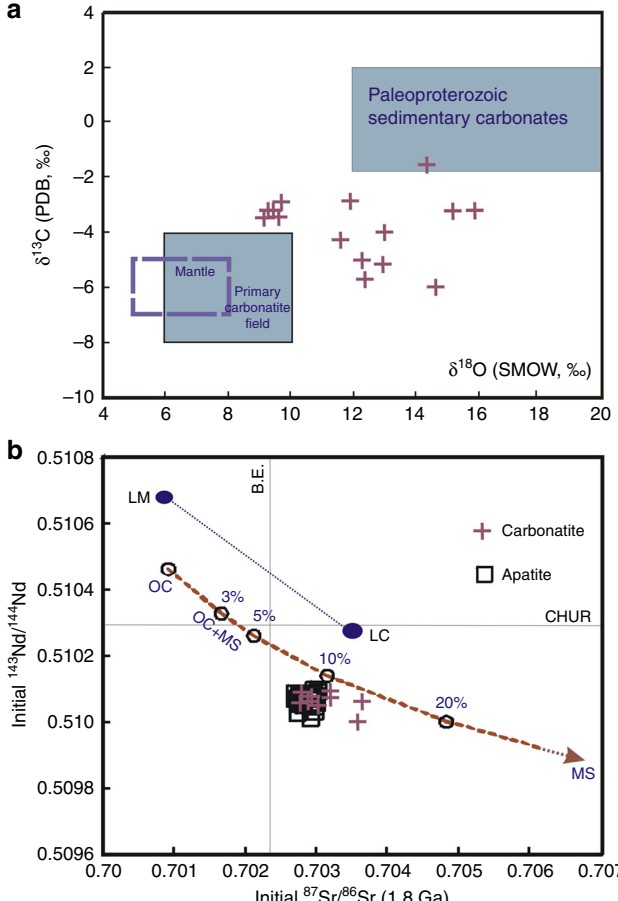

**Fig. 4** Isotopic compositions of the carbonatites. **a** The carbonatite samples from Fengzhen and Huai'an show higher oxygen and carbon isotopes compared to normal mantle-derived rocks (box with dashed line) and typical primary carbonatites (gray box with black line) worldwide. The values of the Paleoproterozoic sedimentary carbonates[30] are also shown to illustrate that heavy-C enrichment in the carbonatites is consistent with the involvement of recycled oceanic carbonates. **b** The initial $^{143}Nd/^{144}Nd$ and $^{87}Rb/^{86}Sr$ ratios of the carbonatites and apatites are compared to the values of the Archean to Proterozoic lithospheric mantle (LM)[31] and the late Archean lower crust in central NCC (LC)[32] with 700-Ma radiogenic addition. The Sr–Nd isotopic variations for the sources of the carbonatites and apatites can be successfully modeled by mixing a 2.5-Ga oceanic crust (OC) with various amounts of 2.5-Ga marine sediments (MS) (Supplementary Table 11). With a single-stage evolution model[54], the isotopic compositions of the two end-members are calculated as ($^{87}Rb/$ $^{86}Sr)_{OC}$ = 0.053 and ($^{147}Sm/^{144}Nd)_{OC}$ = 0.214, and ($^{87}Rb/^{86}Sr)_{MS}$ = 0.186 and ($^{147}Sm/^{144}Nd)_{MS}$ = 0.183, respectively. The initial Sr–Nd isotopic compositions of the LC in the central NCC, and those for the bulk Earth (BE) and chondritic uniform reservoir (CHUR) are calculated for 1.8 Ga

the source. The initial Sr and Nd isotopes of the Archean to Proterozoic lithospheric mantle[31] and TTG crust in the central NCC[32] are very depleted, so that a 700-Ma radiogenic addition cannot produce such an enriched Sr–Nd isotopic signature for the carbonatites. One way to produce the unusually enriched isotopic characteristics of the carbonatites is to add oceanic crust-derived carbonates to a subducted Late Archean (2.5 Ga) basaltic oceanic crust, with a residence time of ~700 Mys. Figure 4b shows the change of the isotopic compositions of carbonatites by mixing a Late Archean oceanic crust with marine sediment. The isotopic composition with an addition of 10% oceanic crust-derived carbonates matches well with our observed values (Fig. 4b),

providing further evidence for the involvement of the oceanic crust in the genesis of the carbonatites.

## Discussion

On the basis of U–Th–Pb dating, compositions of the coexisting minerals in the eclogite xenolith, and isotopic compositions of C, O, Sr, and Nd, we have established that cold deep subduction of ocean crust occurred during the Paleoproterozoic Era in the NCC. The cold subduction environment is also consistent with the preservation of the $Fe^{3+}$-rich majorite inclusion in a websterite-derived garnet in the same carbonatites[27]. The estimated equilibrium temperatures of the host garnet are in the range of 670–1000 °C for depths of 100–200 km, using the garnet–clinopyroxene Fe–Mg-geothermometer[33]. The high $Fe^{3+}$ content ($Fe^{3+}/\Sigma Fe > 0.8$) of the majorite inclusion is likely a product of a redox reaction involving deep carbonatitic magmas, implying that the Paleoproterozoic subduction in the NCC could have been active for an extended period of time. Reduction of carbonate associated with earlier subduction could have provided an environment in which oxidized $Fe^{3+}$-rich majorite formed.

From the regional geology, the low-temperature and high-pressure metamorphism of the carbonated eclogite is the result of the cold Paleoproterozoic oceanic subduction, related to the amalgamation of the Yinshan and Orodos blocks[16]. The 1.91–1.88 Ga Halaqin volcanic belt in Yinshan block, located 100-km to the northwestward of the carbonatite outcrop, includes basalts, andesites, and rhyolites with typical geochemical features of an arc sequence[34]. A recent study also showed that a Paleoproterozoic garnet websterite in the Yinshan block underwent peak metamorphism at ~1.90 Ga when it was subducted to eclogite facies, and then exhumed back to granulite facies at ~1.82 Ga[35]. On a global scale, a semi-continuous record of global kimberlite volcanism starts at ~2 Ga, with a notable increase in kimberlite magmatic activity at 1.2–1.1 Ga[36]. This implies that the cooling ambient mantle directly beneath thermally maturing continental keels could have developed into sites of sustained kimberlite melt generation from 2 Ga. Moreover, low thermal gradient Paleoproterozoic subduction has been recently reported in West African and North American cratons[3,6]. Our work directly links Paleoproterozoic carbonatites to cold slab subduction in the NCC. Figure 5 shows the distribution of Paleoproterozoic carbonatites and Paleoproterozoic granulites and eclogites in orogens worldwide. Multiple experimental studies show that melting of a carbonated slab at pressures greater than 2.5 GPa can form carbonate melt[37,38]. There is a clear association between carbonatites[39] and high-pressure metamorphic rocks in Paleoproterozoic orogens[15].

Although only a few locations of Paleoproterozoic low thermal gradient metamorphism have been reported so far, it is likely that evidence for cold deep subduction in the Paleoproterozoic Era will continue to emerge with expanded exploration to metamorphic terrains, which are intrinsically linked to orogenesis. High-Mg oceanic crust derived from a hotter Archean mantle is less likely to stabilize blueschist-facies mineral assemblages[9]. Moreover, the low-temperature records may be easily overprinted by the followed inter-continental collisions with high-temperature metamorphism worldwide. Therefore, we suggest a common occurrence of modern-style plate tectonics back to at least the Paleoproterozoic Era. The early global-scale subduction could be also associated with the formation of Columbia supercontinent[15].

Global-scale Paleoproterozoic subduction would provide an efficient pathway for surface sediments to enter the ancient deep Earth. Isotopic and trace element geochemical studies of the ocean island basalts have for many years been used to infer the

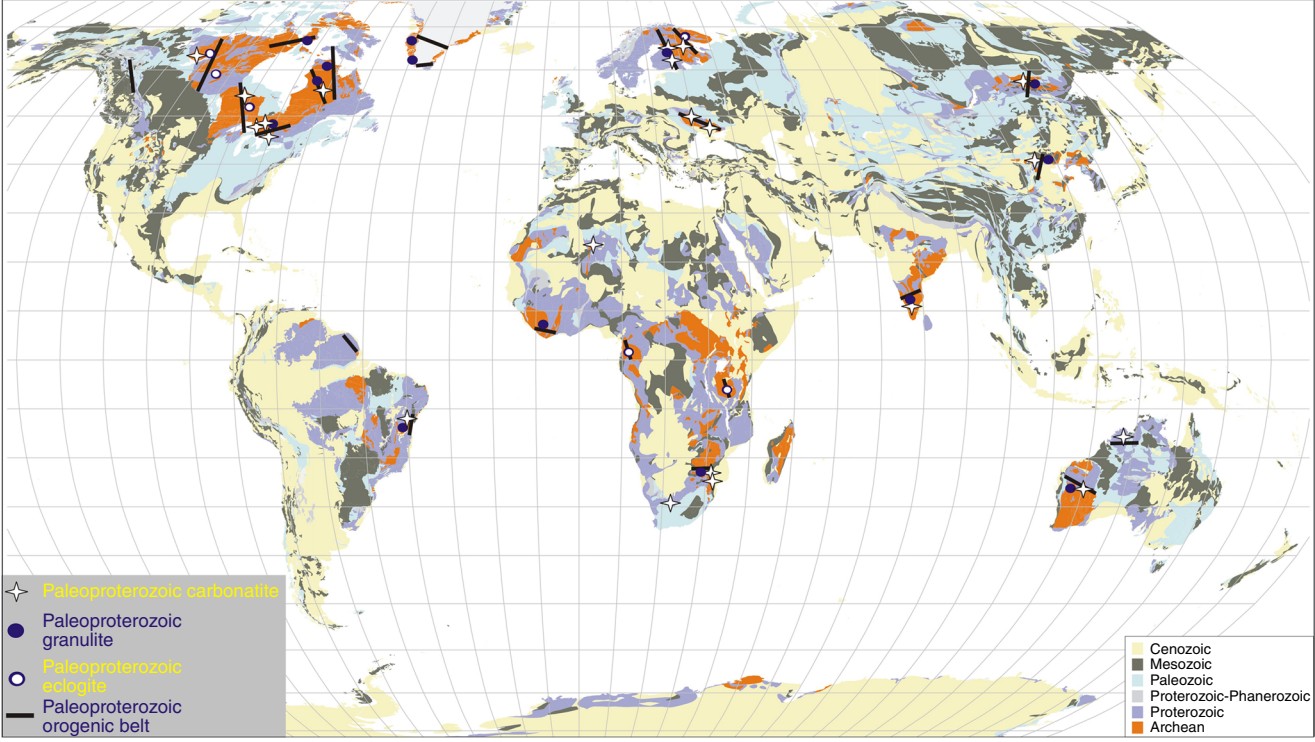

**Fig. 5** Distribution of Paleoproterozoic carbonatites, granulites, and eclogites worldwide. The majority of carbonatite occurrences[39] are associated with high-pressure metamorphism (Supplementary Table 12) in Paleoproterozoic orogens[15]

presence of long-lived (1–2 Ga) compositional heterogeneities in the deep mantle related to recycled oceanic crust together with different types of marine sediments via subduction[40]. However, it is still doubted because of the lack of petrological evidence for ancient deep subduction. Our work not only provides a direct record of deep Paleoproterozoic subduction, but also implies that the recycled carbonates may be played a critical role in the change of local redox state of the deep mantle, including the formation of the $Fe^{3+}$-rich majorite[33] related to the subducted sediments. Melting of the recycled carbonates can also produce melt with enriched trace element and isotope compositions, which are important mantle metasomatic agents that interact with peridotite lithologies[41].

The Paleoproterozoic carbonatites in the Fengzhen and Huai'an areas of the NCC occur in an orogen setting. The eclogite xenoliths hosted by the carbonatites provide a rare window into the ancient deep-subducted slab, showing the first cold subduction in the NCC during the Paleoproterozoic Era and also recording the deepest $Fe^{3+}$-rich majorite inclusion in a cold subduction environment[33]. The characteristics of the Paleoproterozoic slab supports that modern-style plate tectonics has operated since at least the Paleoproterozoic Era. The association of Paleoproterozoic carbonatites with high-pressure metamorphic rocks in Paleoproterozoic orogens worldwide (Fig. 5) argues for sustained, large-scale deep carbon cycle over a long geological period of time, influencing the evolution of mantle oxidation state and compositional heterogeneities in the deep mantle.

## Methods
**Mineral mapping**. The polished eclogite xenolith sample was analyzed using a Tescan Integrated Mineral Analyzer (TIMA) mineralogy system at the Tescan Orsay Holding Lab, Czech Republic. The TIMA comprises a Tescan Mira Schottky field emission scanning electron microscope with four silicon drift energy dispersive (EDS) detectors arranged at 90° intervals around the chamber. The measurements were performed with the dot-mapping mode, and the BSE image was obtained to determine individual particles and boundaries between distinct preliminary phases. A rectangular mesh of measurements on every distinct phase was

obtained with X-ray spectra. The spectroscopic data were matched to mineral definition files, allowing for mineral identification and mapping. The volume and mass ratios of all mineral phases were automatically calculated. The TIMA measurements were performed at 25 kV using a spot size of ~50 nm, a working distance of 15 mm, and a field size set at 1500 μm.

**Apatite dating**. The ages of apatites from the carbonatites were determined by the U–Th–Pb method using a laser (Geolas) ICPMS (Agilent 7500a) at the Institute of Geology and Geophysics, Chinese Academy of Sciences (CAS). The apatites, separated from the carbonatite samples, were casted into resin mounts together with grains of NIST 610, apatite reference materials NW-1 (1160 Ma), and LAP (474 Ma). The beam diameter of the laser is 30 μm with a repetition rate of 8 Hz. The background of Pb and $^{202}$Hg is less than 100 cps. The fractionation correction and results were calculated using the program GLITTER 4.0. The $^{207}$Pb method was applied for common Pb corrections using the upper intercept obtained from a Tera–Wasserburg diagram[42]. The $^{207}$Pb correction method yields more precise and accurate ages than the Tera–Wasserburg Concordia ages because of a small spread in the U/total Pb ratios[42]. The U–Pb analyses of the apatite standards also show that the method of laser ablation can effectively date this mineral without significant matrix effects[43].

**Monazite dating**. The chemical Th–U–total Pb ages of monazite were determined with a wavelength-dispersive X-ray spectrometry (WDS) using a Cameca SX100 electron microprobe at the Masaryk University. Uranium was determined on the U Mβ line (counting time 60 s, detection limit 270 ppm), Thorium on the Th Mα line (counting time 40 s, detection limit 250 ppm), and lead on the Pb Mα line (counting time 240 s, detection limit 130 ppm). Synthetic and natural phases were used as standards, including metallic U for U, PbSe for Pb, CaTh(PO$_4$)$_2$ for Th, synthetic end-member phosphates (XPO$_4$) for REE, Y, sanidine for Si, apatite for Ca, and topaz for F. Data were reduced using the PAP matrix correction routine. Overlapping of peaks and background positions were examined and chosen based on WDS angle scans on natural and synthetic REE-phases. The microprobe was operated at an accelerating voltage of 15 kV and a beam current of 160 nA, with an electron beam defocused to a 2-μm spot to avoid devolatilization, ionic diffusion, and other forms of beam damage on the sample. The monazite age was calculated using the method of refs. 44,45.

**Major element analysis**. The chemical compositions of minerals were obtained with a Cameca SX100 electron microprobe at the Masaryk University, and a JEOL field-emission electron microprobe (JEOL JXA 8530F) at the Geophysical Laboratory of the Carnegie Institution of Washington. The microprobes were operated at an accelerating voltage of 15 kV and a beam current of 30 nA. For each

mineral or a group of minerals, we chose appropriate matrix-specific standards (both natural and synthetic) and optimal instrument conditions (beam settings, detector type, and counting statistics).

**In-situ trace element analysis**. In-situ laser-ablation ICPMS analyses of minerals in the eclogite xenoliths were performed at the School of Earth and Space Sciences, Peking University. The diameter of the ablation spot is 44 μm. NIST 610, 612, and 614 glasses were used as calibration standards for all the samples. The elements used for the internal standards include Ca and Si, expressed as CaO and $SiO_2$ for garnet, omphacite, zoisite, kyanite, and phengite. Analytical error is ≤5% at the ppm level. In-run signal intensity for indicative trace elements was monitored during analysis to make sure that the laser beam stayed within the selected phase and did not penetrate inclusions.

**Trace and major element analysis of rutile**. We used electron probe micro-analyzer (CAMECA SXFive) to analyze eight elements (Si, Fe, Cr, Zr, V, Nb, Ta, and Ti) in rutile at the Institute of Geology and Geophysics, CAS. The microprobe was operated at an accelerating voltage of 20 kV and a beam current of 170 nA. Counting time on the peak of Si, Fe, Cr, Zr, V, Nb, Ta, and Ti is 120, 120, 120, 100, 120, 240, and 10 s, respectively. The rutile R10 was used as standard. The detection limits for the above elements are 23, 35, 32, 35, 12, 54, 69, and 137 ppm, respectively. The eight elements in the standard rutile R10 have been well analyzed by the same electron microprobe[46].

**Isotope analysis**. Carbon and oxygen isotopic compositions of calcite from the carbonatites were measured at the Institute of Geochemistry, CAS, using a continuous-flow isotope ratio mass spectrometer (IsoPrime), with reproducibility of ±0.15‰ and ±0.2‰ for carbon and oxygen, respectively. The results are expressed as per mil (‰) variation relative to PDB for carbon and SMOW for oxygen.

In-situ Sr–Nd isotopes of apatite from the carbonatites were determined by a Neptune LA-MC-ICPMS at the Institute of Geology and Geophysics, CAS. The laser was focused at a height of 60–120 μm above the sample surface and fired using a repetition rate of 6–8 Hz and an energy density of 10 J/cm². The AP1 and AP2 standard apatites were used as external calibrations. Normalized Sr and Nd isotopic ratios were calculated using the exponential law[47].

Sr and Nd isotopic compositions of the carbonatites were analyzed by a MC-ICPMS (VG AXIOM) at Peking University. Mass fractionation corrections for the Sr and Nd isotopic ratios were normalized to $^{86}Sr/^{88}Sr = 0.1194$ and $^{146}Nd/^{144}Nd = 0.7219$, respectively. Repeated measurements for the Nd standard JNdi and Sr standard NBS987 yielded $^{143}Nd/^{144}Nd = 0.512120 \pm 11$ ($2\sigma$) and $^{87}Sr/^{86}Sr = 0.710250 \pm 11$ ($2\sigma$), respectively. To calculate the initial Sr–Nd isotope values, we analyzed the whole-rock Rb, Sr, Sm, and Nd concentrations by solution ICPMS (Thermo Fisher Scientific X-Series II). The data processing procedure includes linear drift correction, internal (matrix) correction, REE and Ba interference corrections, blank subtraction, calibration with international standards, and a dilution correction. Repeat analyses (including well-characterized standards) indicate that the accuracy of trace element measurements is better than 10%.

**P–T pseudosections**. The P–T phase diagram was calculated using THERMO-CALC v. 3.33[27] with an internally consistent thermodynamic data set 5.5[48] (November 2003 upgrade) in the NCKFMASH system. The effective bulk-rock composition is provided in Supplementary Table 5. Water was treated in saturation because of the widespread presence of hydrous phases (such as zoisite and phengite). All Fe was assumed as $Fe^{2+}$. Ti was omitted in the calculations, as rutile is the only Ti-rich phase. Following mineral solid solutions are adopted: garnet[49], clinopyroxene[50], amphibole[51], white mica[52], and chlorite[48].

**Data availability**. The authors declare that the data supporting the findings of this study are available within the paper and its supplementary information files.

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

## Acknowledgments

We thank Antonin Kopřiva and Y. Chen for helping with chemical analysis. We also thank Michael Brown for discussion and suggestions to improve the manuscript. This research was financially supported by the Chinese National Natural Science Foundation (41773022, 41688103, 41520104004), Czechic CEITEC 2020 (LQ1601), and National Science Foundation Geochemistry grant (EAR-1447311) to Y.F.

## Author contributions

C.X., J.K., W.S., and Y.L. were responsible for sample collection. C.X., J.K., Y.F., L.Z., Z.L., M.P., and M.V.G were responsible for mineralogy and petrology. W.S., R.T., Y.Y., and Y.L. performed dating and geochemical analysis. C.X. and Y.F wrote the manuscript with inputs from co-authors.
