## [Peer Review File · Nature Communications]

Reviewers' comments:

Reviewer #1 (Remarks to the Author):

Dear Editor

This is a fascinating and original work and is by all means publishable in Nature Communications. Here Xu and coauthors look at the history of mafic enclaves preserved within carbonatitic magma sampled in the North China Craton to track evidence of high-pressure low-temperature metamorphic conditions ~1.8 Ga ago. They show that low thermal conditions (< 700°C) could developed at the bottom of arcs or collisional belts at the time of the first supercontinent formation, Columbia. This is the first study of it's kind so far as I am aware.

Until the early 2000s, there has been a large consensus in the literature to argue that the first appearance of high-temperature eclogite / high-pressure granulite – type of metamorphism must reflect the initiation of plate subduction (Brown, 2008), in an early style of plate tectonics that became global in late Neoproterozoic time and dominated until the Neoproterozoic (Ernst, 2009; Hoffman, 2006), when modern-style plate tectonics characterized by HP-LT (blueschist) and ultrahigh-P metamorphism began (Stern, 2005).

Data of Xu and coauthors suggest that ultrahigh-P metamorphism already existed in the Paleoproterozoic era, 1 Ga before the expected transitional period. The attached figure (modified after Brown, 2008), given in the supplementary file, illustrates the importance of their findings. The colored text and symbols have been added on the original figure. The red lines delimit the E-HPGM domain (eclogite-high-pressure granulite metamorphism) defined by Brown (2008). This domain is characterized by geothermal gradients of 10-20 °C/Km with temperature up to 600°C.

The red points correspond to geothermal gradients of ~10°C/km identified in Palaeoproterozoic terrains of Siberia (Ota et al, 2004) and North China (Wan et al., 2015), respectively. But it concerns mantle-derived rocks, not crustal samples. Grt-websterites samples from the Shryzhalgai (~2,400 Ga) and Alxa (~1,900 Ga) complexes in the North China Craton yielded pressure of 2.5 Gpa. They probably originate from a lithospheric mantle section metasomatized by fluids at ~100 km depth before to be exhumed, supporting the conclusion of authors that UHP conditions can be generated in arc setting. However, temperature associated with the peak of pressure is much more elevated (> 350°C) than the peak of temperature estimated by authors for crust-derived eclogitic enclaves in carbonatites.

Whether the authors' ultimate arguments are valid or not (do the enclaves really correspond to crumbled pieces of oceanic slab dragged to the surface by magma in arc or collisional settings? underplating of oceanic crust within a cold lithospheric mantle could be also considered, accounting for a low thermal regime at that time; Tappe et al., 2016, EPSL), their approach is very original and their ideas and hypotheses are quite interesting and worth bringing to the attention of a wide audience.

I think this result will be also influential, in causing/ pushing other scientists to look at similar lava to track evidence of ultra-high pressure (UHP) metamorphic conditions in the Precambrian record. This is clearly a change of paradigm in our vision and way to investigate the thermal evolution of Precambrian belts. For a long time, the geosciences community looked after blueschists that were considered as the hallmark of modern-style plate subductions, involving oceanic crust. However, recent theoretical calculations shown that Mg-rich composition (in average) of Precambrian oceanic crust should hamper the genesis of blue amphiboles in metamorphic assemblages (Palin & White, 2016). The dramatic consequence is that we were looking for metamorphic evidence that could not form in subduction domains at that time. That is evidently a dead end road leading to cyclical reasoning. Xu et coauthors's results will have the same electroshock effect. We had a tendency to limit our exploration to metamorphic terrains, whereas it should expand within magmatic domains

which are intrinsically linked to Precambrian orogenesis.

There is thus an urgent need to look for other metamorphic assemblages than blueschists and more critically to enlarge our investigations to magmatic terrains (that obviously can preserve UHP relics!)

I only have few moderate suggestions, neither of which should impede publication.:

(1) Could you discuss a little bit more in detail the geodynamic evolution of the North China Craton in the Palaeoproterozoic ? It will help the reader to better assess the temporal and spatial links between metamorphic and magmatic events ;

(2) Could you also discuss the relationships between (i) secular cooling of the mantle, (ii) the extremely low temperatures evidenced at 100 km depth below your Palaeoproterozoic arc and (iii) the production of carbonatite at that time (that classically occur in rift setting or remnant arc settings, like in Iran). You could find some interesting "foods" in the recent paper of Tappe et al (2016) (Geodynamics of kimberlites on a cooling Earth: Clues to plate tectonic evolution and deep volatile cycles) ;

(3) High-pressure mantle-derived enclaves brought to surface by Phanerozoic magma (kimberlite) exist in the literature and their PT record is classically used to derive continental geotherms in Precambrian cratons. You could compare your PT results (2.5 Gpa, 650°C) with those of authors (Rudnick & Nyblade, 1999) as they plot at the extreme side of their curves, that is very interesting I think...

Jérôme Ganne

Reviewer #2 (Remarks to the Author):

This paper presents pressure-temperature estimates from an eclogite xenolith in a 1.8 Ga carbonatite to argue for the onset of modern-style plate tectonics in the Palaeoproterozoic. In addition, isotopic data from the host carbonatite are used to argue that subducted carbonate material was involved in the generation of the magma, suggesting that a deep carbon cycle had also already developed by this time.

The first conclusion, that the eclogite xenolith indicates the onset of modern-style plate tectonics (even though it is not specifically defined what is meant by modern-style plate tectonics) cannot be conclusively supported by the data presented because of the setting of the eclogite as a xenolith in carbonatite. The two other examples of Palaeoproterozoic cold subduction cited here (the blueschists from West Africa and the eclogites from the Trans-Hudson Orogen) are both from tectonic units that are demonstrably related to the orogen they are found in. This is crucial, because it indicates that the rocks were exhumed by the same structure that subducted them in the first place. The exhumation aspect is key, because it indicates that the subduction zone extended continuously from the surface through blueschist and eclogite facies conditions, providing direct evidence for long-lived, continuous deep subduction. Without the exhumation context, all that this sample proves is that material was recycled from the surface to the mantle. This is not novel, as eclogite xenoliths derived from oceanic crust as old as 3 Ga have been reported from a number of kimberlite localities worldwide (see reviews by Jacob, 2004, *Lithos* 77, 295-316 and Aulbach & Jacob, 2016, *Lithos* 262, 586-605; Shirey & Richardson 2011, *Science* 333, 434-436; various recent papers by S. Tappe, K. Smart and co-authors as a starting point). These occurrences indicate the (at least intermittent) operation of subduction since the Mesoproterozoic, as has been modelled by Van Hunen & Van den Berg 2008 *Lithos* 103, 217-235 and Sizova et al,

2010 *Lithos* 116, 209-229. The occurrence described here does not contribute anything new to this discussion, and does not indicate the initiation of modern-style plate tectonics as claimed.

Curiously, only the major element composition of the eclogite is used to indicate a MORB protolith, whereas all the isotopic studies are from the host carbonatite. Why are no stable isotope of trace element patterns presented that might indicate low-T seafloor processes in the protolith?

I am not an expert on the origin of carbonatite magmas, but the isotopic signature of this carbonatite is not unusual, and its interpretation as indicative of a surface carbonate origin is not as unequivocal as implied by the authors and the single reference they cite in support. Refer to e.g. the review by Deines, 2002, *Earth Science Reviews* 58, 247-278 for caveats with regards to the interpretation of carbon isotope data from mantle rocks, which would provide a starting point for a more balanced discussion of these data.

Reviewer #3 (Remarks to the Author):

This manuscript reports highly significant findings that support a recent and important direction in thinking in the community that the absence of high-pressure/low-temperature rocks older than c. 0.8 Ga in the geological record does not necessarily indicate a lack of modern-day style subduction at that time. An increasing amount of evidence for cold subduction having operated in the Proterozoic is coming to light every year, and this contribution marks the first reported examples from China. The implications of this result for global tectonics and the long-term carbon cycle are profound and well discussed. One qualm is that I would like to see more descriptions and photographs of the field-/outcrop-scale features of these rocks, and perhaps additional photomicrographs, as the petrological descriptions could be expanded upon. This material is not necessary for the main manuscript but would be a welcome addition to supplementary information. Otherwise, I have made several suggestions below for changes that can be made to the main text in order to improve clarity and listed key references that have made recent and significant contributions to this plate tectonic debate, which I would hope to see recognized in the revised version.

Lines 26–28: This is one explanation, but it is not the only one and perhaps not even the most popular one. It is becoming more and more accepted that this signature of “missing” high-P/low-T rocks prior to 0.8 Ga is simply a sampling issue and not representative of the absence of subduction (see Korenaga, 2016). In fact, as you summarize in your own work, multiple recent studies have revealed an increasing amount of evidence for cold subduction dating back as far as the Paleoproterozoic. A lack of preservation of old blueschists/eclogites may be to blame, as it is difficult to preserve minerals such as lawsonite, even in modern-day tectonic environments (e.g. Whitney and Davis, 2006; *Geology*). Alternatively, the more mafic (Mg-rich) oceanic crust in the Archean and Proterozoic compared to modern-day MORB has been shown by Palin and White (2016) *Nature Geoscience* to have precluded diagnostic blueschist-facies minerals from forming, meaning that cold subduction could have operated but we are not immediately recognizing the rocks as having formed at those conditions. These alternative theories should be acknowledged here also.

- Korenaga, J., 2016. Plate tectonics: Metamorphic myth. *Nature Geoscience*, v. 9, p. 9.
- Whitney, D.L. and Davis, P.B., 2006. Why is lawsonite eclogite so rare? Metamorphism and preservation of lawsonite eclogite, Sivrihisar, Turkey. *Geology*, v. 34, p. 473–476.
- Palin, R.M., and White, R.W., 2016. Emergence of blueschists on Earth linked to secular changes in oceanic crust composition. *Nature Geoscience*, v. 9, p. 60–64

Lines 49–52: This referenced study (no. 7), cited on lines 51 and 52, did not examine the viability of subduction in the Archean; they examined how metamorphism can cause density changes at the base of overthickened crust. Please replace with a more suitable reference (e.g. Fischer and Gerya, 2016)

- Fischer, R. and Gerya, T., 2016. Regimes of subduction and lithospheric dynamics in the Precambrian: 3D thermomechanical modelling. *Gondwana Research*, v. 37, p. 53–70.

Lines 52–54: There are petrological arguments for shallow subduction also, including those that state that trace element signatures in Archean TTG magmas can only be produced when melts separate from a garnet amphibolite residue, not an eclogite residue. Thermodynamic modeling of metamorphic mineral assemblages has shown that this can only occur during shallow subduction. Please add this detail here, as the petrological record is a more appropriate source of evidence than geodynamic models, which necessarily must make many assumptions that are not always justified.

- Moyaen, J.F. and Martin, H., 2012. Forty years of TTG research. *Lithos*, v. 148, p. 312–336.
- Palin, R.M., White, R.W. and Green, E.C., 2016. Partial melting of metabasic rocks and the generation of tonalitic–trondhjemitic–granodioritic (TTG) crust in the Archaean: constraints from phase equilibrium modelling. *Precambrian Research*, v. 287, p. 73–90.

Lines 95–96: There three points here. Firstly, I wonder how such precision (three decimal places) can be obtained via either the energy dispersive and wavelength dispersive analytical systems described in the Methods? Even with the latter, compositions are likely no more accurate than 2 decimal places. Secondly, as garnet may contain Fe³⁺ in addition to Fe²⁺, this will affect your total cation counts: how did you examine the ferric/ferrous iron ratio? Thirdly, as a stoichiometric garnet should have 3.00 Si cations per formula unit for 12 oxygens, is 3.02 Si really elevated in any way?

Line 102: Replace “clear” with “inclusion-free”

Lines 142–143: Is these zircon in your rocks? I did not see it listed as an accessory phase. If it is not present, temperatures obtained from this thermometer will represent minima and may be notably lower than the true peak temperature (i.e. the system is not saturated in Zr).

Lines 144–147: What are the uncertainties on these numbers? The Zack et al. (2004) paper should provide these data. They should be listed alongside the values themselves.

Lines 263–267: Yes, these caveats should be emphasized with the references given earlier, in addition to the petrological modeling of Palin and White (2016) showing that higher-Mg crust derived from a hotter Archean mantle is less likely to stabilize blueschist-facies mineral assemblages. These are important points that have helped to slowly deconstruct the outdated hypothesis that subduction is confined to the Neoproterozoic.

- Palin, R.M., and White, R.W., 2016. Emergence of blueschists on Earth linked to secular changes in oceanic crust composition. *Nature Geoscience*, v. 9, p. 60–64

Line 396: I assume that the NSF grant number should be provided here.

Lines 496–504: Alongside the uncertainties that are associated with the Zr-in-rutile thermometer and the Grt-Cpx-Phen barometer, there are also uncertainties associated with the positions of phase boundaries on your calculated pseudosection, which themselves can be significant. Monte Carlo modeling suggests that errors can reach ± 1 kbar (cf. Powell and Holland, 2008; Palin et al., 2016), which should be noted and considered, as they will introduce further uncertainty into your calculated geotherm. As your interpretation of metamorphism being along a cold geotherm is critical to your overall conclusions, all key sources of uncertainty should be described and examined so that the reader can assess the veracity of your results.

- Powell, R. and Holland, T.J.B., 2008. On thermobarometry. *Journal of Metamorphic Geology*, v. 26, p. 155–179.

- Palin, R.M., Weller, O.M., Waters, D.J. and Dyck, B., 2016. Quantifying geological uncertainty in metamorphic phase equilibria modelling; a Monte Carlo assessment and implications for tectonic interpretations. *Geoscience Frontiers*, v. 7, p. 591–607.

Point-by-point responses to reviewers' comments:

Reviewer #1:

1. *This is a fascinating and original work and is by all means publishable in Nature Communications. Here Xu and coauthors look at the history of mafic enclaves preserved within carbonatitic magma sampled in the North China Craton to track evidence of high-pressure low-temperature metamorphic conditions ~1.8 Ga ago. They show that low thermal conditions (< 700°C) could developed at the bottom of arcs or collisional belts at the time of the first supercontinent formation, Columbia. This is the first study of it's kind so far as I am aware.*

Reply: We thank Dr. Jérôme Ganne for his encouraging word and support.

2. *Until the early 2000s, there has been a large consensus in the literature to argue that the first appearance of high-temperature eclogite / high-pressure granulite – type of metamorphism must reflect the initiation of plate subduction (Brown, 2008), in an early style of plate tectonics that became global in late Neoproterozoic time and dominated until the Neoproterozoic (Ernst, 2009; Hoffman, 2006), when modern-style plate tectonics characterized by HP-LT (blueschist) and ultrahigh-P metamorphism began (Stern, 2005).*

Data of Xu and coauthors suggest that ultrahigh-P metamorphism already existed in the Paleoproterozoic era, 1 Ga before the expected transitional period. The attached figure (modified after Brown, 2008), given in the supplementary file, illustrates the importance of their findings. The colored text and symbols have been added on the original figure. The red lines delimit the E-HPGM domain (eclogite-high-pressure granulite metamorphism) defined by Brown (2008). This domain is characterized by geothermal gradients of 10-20 °C/Km with temperature up to 600°C. The red points correspond to geothermal gradients of ~10°C/km identified in Palaeoproterozoic terrains of Siberia (Ota et al, 2004) and North China (Wan et al., 2015), respectively. But it concerns mantle-derived rocks, not crustal samples. Grt-websterites samples from the Shryzhalgai (~2,400 Ga) and Alxa (~1,900 Ga) complexes in the North China Craton yielded pressure of 2.5 Gpa. They probably originate from a lithospheric mantle section metasomatized by fluids at ~100 km depth before to be exhumed, supporting the conclusion of authors that UHP conditions can be generated in arc setting. However, temperature associated with the peak of pressure is much more elevated (> 350°C) than the peak of temperature estimated by authors for crust-derived eclogitic enclaves in carbonatites.

Whether the authors' ultimate arguments are valid or not (do the enclaves really correspond to crumbled pieces of oceanic slab dragged to the surface by magma in arc or collisional settings? underplating of oceanic crust within a cold lithospheric mantle could be also considered, accounting for a low thermal regime at that time; Tappe et al., 2016, EPSL), their approach is very original and their ideas and hypotheses are quite interesting and worth bringing to the attention of a wide audience.

I think this result will be also influential, in causing/ pushing other scientists to look at similar lava to track evidence of ultra-high pressure (UHP) metamorphic conditions in the Precambrian record. This is clearly a change of paradigm in our vision and way to investigate the thermal evolution of Precambrian belts. For a long time, the geosciences community looked after blueschists that were considered as the hallmark of modern-style plate subductions, involving oceanic crust. However, recent theoretical calculations shown that Mg-rich composition (in average) of Precambrian oceanic crust should hamper the genesis of blue amphiboles in metamorphic assemblages (Palin & White, 2016). The dramatic consequence is that we were looking for metamorphic evidence that could not form in subduction domains at that time. That is evidently a dead end road leading to cyclical reasoning. Xu et coauthors' results will have the same electroshock effect. We had a tendency to limit our exploration to metamorphic terrains, whereas it should expand within magmatic domains which are intrinsically linked to Precambrian orogenesis.

There is thus an urgent need to look for other metamorphic assemblages than blueschists and more critically to enlarge our investigations to magmatic terrains (that obviously can preserve UHP relics!).

Reply: Thank you for the insightful comments. We agree that the blueschists are not the only mark for modern-style plate tectonics. Palin and White (*Nat. Geosci.*, 2016, 9, 60–64) gave strong evidence that Mg-rich composition of Precambrian oceanic crusts may preclude the blueschist formation. This work has been cited in the revision. We also cited the work by Wan et al. (*Nat. Commun.* 2015, 6, 8344) and Tappe et al. (*Earth Planet. Sci. Lett.* 2018, 484, 1-14) to support the onset of modern-style plate tectonics worldwide at the Paleoproterozoic Era.

3. *I only have few moderate suggestions, neither of which should impede publication:*
(1) Could you discuss a little bit more in detail the geodynamic evolution of the North China Craton in the Palaeoproterozoic? It will help the reader to better assess the temporal and spatial links between metamorphic and magmatic events.

Reply: We highlighted the geological background in Result along with a new figure (Figure 1) that shows the geological map of North China craton and field photos of carbonatite, high-pressure granulite, and eclogite. Additional information is provided in supplementary figure 1. Our carbonatites and the hosted eclogite xenoliths occur in the Paleoproterozoic Trans–North China Orogen. Many high-pressure granulites are distributed in the orogen, and show similar metamorphic ages as the eclogite.

4. *Could you also discuss the relationships between (i) secular cooling of the mantle, (ii) the extremely low temperatures evidenced at 100 km depth below your Palaeoproterozoic arc and (iii) the production of carbonatite at that time (that classically occur in rift setting or remnant arc settings, like in Iran). You could find*

some interesting “foods” in the recent paper of Tappe et al (2016) (Geodynamics of kimberlites on a cooling Earth: Clues to plate tectonic evolution and deep volatile cycles).

Reply: We cited the work by Tappe et al. (2018) to discuss that cooling ambient mantle directly beneath thermally maturing continental keels develops into sites of sustained kimberlite melt generation from 2 Ga. A semi-continuous record of global kimberlite volcanism starts at ~2 Ga, implying that the Paleoproterozoic mantle in more than 200 km depth began to cool to less 1400 °C to form kimberlite magmas, which replaced high-temperature komatiites.

5. *High-pressure mantle-derived enclaves bring to surface by Phanerozoic magma (kimberlite) exist in the literature and their PT record is classically used to derive continental geotherms in Precambrian cratons. You could compare your PT results (2.5 Gpa, 650°C) with those of authors (Rudnick & Nyblade, 1999) as they plot at the extreme side of their curves, that is very interesting I think.*

Reply: Our eclogite xenoliths have different mineral and chemical compositions comparing to the mantle eclogite xenoliths captured by Phanerozoic kimberlites, which has been discussed in the revision and shown in Supplementary Figure 6.

Reviewer #2:

1. *The first conclusion, that the eclogite xenolith indicates the onset of modern-style plate tectonics (even though it is not specifically defined what is meant by modern-style plate tectonics) cannot be conclusively supported by the data presented because of the setting of the eclogite as a xenolith in carbonatite. The two other examples of Palaeoproterozoic cold subduction cited here (the blueschists from West Africa and the eclogites from the Trans-Hudson Orogen) are both from tectonic units that are demonstrably related to the orogen they are found in. This is crucial, because it indicates that the rocks were exhumed by the same structure that subducted them in the first place. The exhumation aspect is key, because it indicates that the subduction zone extended continuously from the surface through blueschist and eclogite facies conditions, providing direct evidence for long-lived, continuous deep subduction. Without the exhumation context, all that this sample proves is that material was recycled from the surface to the mantle. This is not novel, as eclogite xenoliths derived from oceanic crust as old as 3 Ga have been reported from a number of kimberlite localities worldwide (see reviews by Jacob, 2004, Lithos 77, 295-316 and Aulbach & Jacob, 2016, Lithos 262, 586-605; Shirey & Richardson 2011, Science 333, 434-436; various recent papers by S. Tappe, K. Smart and co-authors as a starting point). These occurrences indicate the (at least intermittent) operation of subduction since the Mesoarchaeon, as has been modelled by Van Hunen & Van den Berg 2008 Lithos 103, 217-235 and Sizova et al, 2010 Lithos 116, 209-229. The occurrence described here does not contribute anything new to this discussion, and does not indicate the initiation of modern-style plate tectonics as claimed.*

Reply: The reviewer seems to be skeptical about the association of our eclogite with an orogeny. We added detailed geological background to emphasize our carbonatite and eclogite xenolith occurrence in the Trans-North China Orogen. Many Paleoproterozoic high-pressure granulites have been found in the orogenic belt. Their pressure-temperature paths are mostly clockwise, related to collision/exhumation processes, and some of them evolved through eclogite facies. The carbonatites are also associated with the granulites (see supplementary figure 1). The latter has similar metamorphic ages as the eclogites. Almost all eclogites may have been decomposed to high-pressure granulites during exhumation. However, the carbonatite magma has extremely low viscosity that facilitates eclogite preservation by its rapid ascent to the surface.

Our eclogite xenoliths have different mineral and chemical compositions compared to the mantle eclogite xenoliths in kimberlites (see supplementary figure 6, and Jacob, *Lithos*, 2004, 77, 295-316). The origin of the mantle eclogite xenoliths in kimberlites is still disputed, including high-pressure cumulates from mantle melts or a 'shallow' oceanic lithosphere origin (see Smart et al., *Earth Planet. Sci. Lett.* 2012, 319-320, 165-177). It is possible that the operation of subduction started since the Mesoarchean, but the subduction is shallow and it is distinguished from modern deep cold subduction. Many 2.5-3 Ga komatiites have been found in the Archean cratons, indicating the Archean mantle temperature is quite high. Therefore modern-style deep subduction would be precluded.

Our work not only supports cold subduction in the Paleoproterozoic Era, but also gives direct petrological evidence for carbonatite origin related to the subducted slab in orogen. We further show that many Paleoproterozoic carbonatites worldwide are associated with granulites and eclogites in Paleoproterozoic orogenic belts. Therefore, we suggest that the modern-style plate tectonics have operated since the Paleoproterozoic Era to form the first supercontinent of Columbia, and to cause large-scale deep carbon cycle over a long geological period that could influence the evolution of the mantle oxidation state and compositional heterogeneities in the deep mantle.

2. *Curiously, only the major element composition of the eclogite is used to indicate a MORB protolith, whereas all the isotopic studies are from the host carbonatite. Why are no stable isotope of trace element patterns presented that might indicate low-T seafloor processes in the protolith?*

Reply: We added additional data on trace element compositions and a discussion. The eclogite xenoliths are small, rare, and contaminated by REE-rich calcites, so they are not good for whole rock analysis. The trace element compositions of garnet, omphacite, zoisite, phengite, and kyanite have been analyzed by in-situ laser-ablation ICPMS, shown in new table and figure (Supplementary Table 6 and Supplementary Figure 7). According to the major element compositions for komatiites, oceanic gabbros, and MORB

(Jacob, Lithos, 2004, 77, 295-316), the eclogite plots in the oceanic gabbro field (see new Supplementary Figure 6). Using the mineral compositions and modal mass, we reconstructed the REE pattern for eclogite that shows slightly depleted LREEs ($\text{La/Yb}_{\text{cn}} = 0.7$), flat HREE pattern ($\text{Y/Yb}_{\text{cn}} = 1$), and positive Eu anomaly. The main hosts for incompatible trace elements in the eclogite are zoisite [LREEs, Sr (3574-3988 ppm), Th (2-10 ppm), U (1-2 ppm), Pb (10-11 ppm)], garnet [Y (13-31 ppm), HREEs], and phengite [Rb (183-201 ppm), Ba (2428-2564)]. This is consistent with the study of Korh et al. (J. Petrol., 2009, 50, 1107-1148) that the trace-element budgets in eclogites are defined by newly formed minerals of zoisite, garnet, and phengite during prograde metamorphism in the subduction zones.

3. *I am not an expert on the origin of carbonatite magmas, but the isotopic signature of this carbonatite is not unusual, and its interpretation as indicative of a surface carbonate origin is not as unequivocal as implied by the authors and the single reference they cite in support. Refer to e.g. the review by Deines, 2002, Earth Science Reviews 58, 247-278 for caveats with regards to the interpretation of carbon isotope data from mantle rocks, which would provide a starting point for a more balanced discussion of these data.*

Reply: Deines (2002) discussed the carbon isotope geochemistry of mantle xenoliths. It is normal that many mantle xenoliths show bimodal with peaks at -5‰ and -25‰, reflecting mantle degassing and recycled inorganic carbon. In contrast, our carbonatites show C isotope enrichment, and the data plot towards the field of Paleoproterozoic sedimentary carbonates (Fig. 4). They are also different from typical cratonic carbonatites. The possible factors for high $\delta^{13}\text{C}$ values have been discussed in the manuscript. The involvement of subducted oceanic carbonates is our preferred model, which is also supported by enriched Sr-Nd isotope composition. The carbonatites are characterized by very unusual negative $\epsilon_{\text{Nd}}(t)$ value and high $^{87}\text{Sr}/^{86}\text{Sr}$, distinguished them from the lithospheric mantle and lower continental crust compositions. A model of mixing oceanic crust with marine sediment has been shown in Figure 4b.

Reviewer #3:

1. *This manuscript reports highly significant findings that support a recent and important direction in thinking in the community that the absence of high-pressure/low-temperature rocks older than c. 0.8 Ga in the geological record does not necessarily indicate a lack of modern-day style subduction at that time. An increasing amount of evidence for cold subduction having operated in the Proterozoic is coming to light every year, and this contribution marks the first reported examples from China. The implications of this result for global tectonics and the long-term carbon cycle are profound and well discussed. One qualm is that I would like to see more descriptions and photographs of the field-/outcrop-scale features of these rocks, and perhaps additional photomicrographs, as the petrological descriptions could be expanded upon. This*

material is not necessary for the main manuscript but would be a welcome addition to supplementary information. Otherwise, I have made several suggestions below for changes that can be made to the main text in order to improve clarity and listed key references that have made recent and significant contributions to this plate tectonic debate, which I would hope to see recognized in the revised version.

Reply: Thanks for positive feedback and suggestions. More descriptions of geological setting have been added in the Result section. Field photographs of the carbonatites, granulite, and eclogite have been added as supplementary figures.

- 2. Lines 26–28: This is one explanation, but it is not the only one and perhaps not even the most popular one. It is becoming more and more accepted that this signature of “missing” high-P/low-T rocks prior to 0.8 Ga is simply a sampling issue and not representative of the absence of subduction (see Korenaga, 2016). In fact, as you summarize in your own work, multiple recent studies have revealed an increasing amount of evidence for cold subduction dating back as far as the Paleoproterozoic. A lack of preservation of old blueschists/eclogites may be to blame, as it is difficult to preserve minerals such as lawsonite, even in modern-day tectonic environments (e.g. Whitney and Davis, 2006; Geology). Alternatively, the more mafic (Mg-rich) oceanic crust in the Archean and Proterozoic compared to modern-day MORB has been shown by Palin and White (2016) Nature Geoscience to have precluded diagnostic blueschist-facies minerals from forming, meaning that cold subduction could have operated but we are not immediately recognizing the rocks as having formed at those conditions. These alternative theories should be acknowledged here also.*

Reply: We agree. The abstract has been rewritten to reflect the comment.

- 3. Lines 49–52: This referenced study (no. 7), cited on lines 51 and 52, did not examine the viability of subduction in the Archean; they examined how metamorphism can cause density changes at the base of overthickened crust. Please replace with a more suitable reference (e.g. Fischer and Gerya, 2016).*

Reply: The reference of Fischer and Gerya (2016) is cited in the revision.

- 4. Lines 52–54: There are petrological arguments for shallow subduction also, including those that state that trace element signatures in Archean TTG magmas can only be produced when melts separate from a garnet amphibolite residue, not an eclogite residue. Thermodynamic modeling of metamorphic mineral assemblages has shown that this can only occur during shallow subduction. Please add this detail here, as the petrological record is a more appropriate source of evidence than geodynamic models, which necessarily must make many assumptions that are not always justified.*

Reply: We include the works of Moyen and Martin (2012) and Palin et al. (2016) in the revision.

5. *Lines 95–96: There three points here. Firstly, I wonder how such precision (three decimal places) can be obtained via either the energy dispersive and wavelength dispersive analytical systems described in the Methods? Even with the latter, compositions are likely no more accurate than 2 decimal places. Secondly, as garnet may contain Fe³⁺ in addition to Fe²⁺, this will affect your total cation counts: how did you examine the ferric/ferrous iron ratio? Thirdly, as a stoichiometric garnet should have 3.00 Si cations per formula unit for 12 oxygens, is 3.02 Si really elevated in any way?*

Reply: The errors in chemical compositions represent standard deviations from microprobe analysis. Cation numbers were calculated on the basis of 12 oxygen atoms, with their errors propagated from the original measurement uncertainties. The Fe³⁺ content in garnet was estimated from charge balance, but we cannot rule out a small amount of Fe³⁺ in the garnet. We are not claiming the elevated Si in garnet is due to the majoritic component as other assumption in the formula calculations could affect the total cation counts as suggested by the reviewer.

6. *Line 102: Replace “clear” with “inclusion-free”*

Reply: It is replaced.

7. *Lines 142–143: Is these zircon in your rocks? I did not see it listed as an accessory phase. If it is not present, temperatures obtained from this thermometer will represent minima and may be notably lower than the true peak temperature (i.e. the system is not saturated in Zr).*

Reply: Zircon inclusions have been observed in the eclogitic garnet. We added a supplementary figure (Supplementary Figure 5) as an example.

8. *Lines 144–147: What are the uncertainties on these numbers? The Zack et al. (2004) paper should provide these data. They should be listed alongside the values themselves.*

Reply: We listed the uncertainties in supplementary table.

9. *Lines 263–267: Yes, these caveats should be emphasized with the references given earlier, in addition to the petrological modeling of Palin and White (2016) showing that higher-Mg crust derived from a hotter Archean mantle is less likely to stabilize blueschist-facies mineral assemblages. These are important points that have helped to slowly deconstruct the outdated hypothesis that subduction is confined to the Neoproterozoic.*

Reply: Yes, we included the work of Palin and white (2016), which provides strong evidence that lack of old blueschist cannot be used as index to confine the modern-style subduction at the Neoproterozoic.

10. *Line 396: I assume that the NSF grant number should be provided here.*

Reply: The grant number is added.

11. *Lines 496–504: Alongside the uncertainties that are associated with the Zr-in-rutile thermometer and the Grt-Cpx-Phen barometer, there are also uncertainties associated with the positions of phase boundaries on your calculated pseudosection, which themselves can be significant. Monte Carlo modeling suggests that errors can reach ± 1 kbar (cf. Powell and Holland, 2008; Palin et al., 2016), which should be noted and considered, as they will introduce further uncertainty into your calculated geotherm. As your interpretation of metamorphism being along a cold geotherm is critical to your overall conclusions, all key sources of uncertainty should be described and examined so that the reader can assess the veracity of your results.*

Reply: In addition to the uncertainties discussed in the text, we also added the statement that the standard deviation (± 1.4 kbar) in pressure is adopted according to a Monte Carlo assessment on the phase equilibrium modeling (Palin et al., 2016).